# Identification of the Dominant Factors in Groundwater Recharge Process, Using Multivariate Statistical Approaches in a Semi-Arid Region

**José Luis Uc Castillo** [1], **José Alfredo Ramos Leal** [1], **Diego Armando Martínez Cruz** [2], **Adrián Cervantes Martínez** [3] **and Ana Elizabeth Marín Celestino** [4,*]

1 Instituto Potosino de Investigación Científica y Tecnológica, A.C. División de Geociencias Aplicadas, Camino a la Presa San José 2055, Lomas 4a Sección, San Luis Potosí 78216, Mexico; luis79505@gmail.com (J.L.U.C.); jalfredo@ipicyt.edu.mx (J.A.R.L.)

2 CONACYT-Centro de Investigación en Materiales Avanzados, S.C. Calle CIMAV 110, Ejido Arroyo Seco, Col. 15 de mayo (Tapias), Durango 34147, Mexico; diego.martinez@cimav.edu.mx

3 Unidad Académica Cozumel, Universidad de Quintana Roo, Av. Andrés Quintana Roo, Calle 11 con calle 110 sur s/n, Cozumel 77600, Mexico; adcervantes@uqroo.edu.mx

4 CONACYT-Instituto Potosino de Investigación Científica y Tecnológica, A.C. División de Geociencias Aplicadas, Camino a la Presa San José 2055, Col. Lomas 4ta Sección, San Luis Potosí 78216, Mexico

* Correspondence: ana.marin@ipicyt.edu.mx; Tel.: +52-444-834-2000

**Abstract:** Identifying contributing factors of potential recharge zones is essential for sustainable groundwater resources management in arid regions. In this study, a data matrix with 66 observations of climatic, hydrogeological, morphological, and land use variables was analyzed. The dominant factors in groundwater recharge process and potential recharge zones were evaluated using *K-means* clustering, principal component analysis (PCA), and geostatistical analysis. The study highlights the importance of multivariate methods coupled with geospatial analysis to identify the main factors contributing to recharge processes and delineate potential groundwater recharge areas. Potential recharge zones were defined into cluster 1 and cluster 3; these were classified as low potential for recharge. Cluster 2 was classified with high potential for groundwater recharge. Cluster 1 is located on a flat land surface with nearby faults and it is mostly composed of ignimbrites and volcanic rocks of low hydraulic conductivity (K). Cluster 2 is located on a flat lowland agricultural area, and it is mainly composed of alluvium that contributes to a higher hydraulic conductivity. Cluster 3 is located on steep slopes with nearby faults and is formed of rhyolite and ignimbrite with interbedded layers of volcanic rocks of low hydraulic conductivity. PCA disclosed that groundwater recharge processes are controlled by geology, K, temperature, precipitation, potential evapotranspiration (PET), humidity, and land use. Infiltration processes are restricted by low hydraulic conductivity, as well as ignimbrites and volcanic rocks of low porosity. This study demonstrates that given the climatic and geological conditions found in the Sierra de San Miguelito Volcanic Complex (SSMVC), this region is not working optimally as a water recharge zone towards the deep aquifer of the San Luis Potosí Valley (SLPV). This methodology will be useful for water resource managers to develop strategies to identify and define priority recharge areas with greater certainty.

**Keywords:** groundwater recharge; infiltration; *K-means* clustering; PCA; Sierra de San Miguelito Volcanic Complex

## 1. Introduction

Groundwater recharge is an essential part of the hydrological cycle and an essential factor for the sustainable management of hydric resources [1–4]. Recharge feeds aquifers and is essential to balance the demand and supply of water to develop activities, particularly in arid and semi-arid regions [5–7]. In those regions, surface water is scanty, so groundwater is a safe resource to meet water supply needs of populations. Increased

groundwater consumption has put pressure on natural recharge, altering the balance between recharge and discharge into aquifers [1,8,9]. The groundwater recharge process by precipitation in arid and semi-arid zones is highly variable in space due to the extreme climate (e.g., very high temperatures and evapotranspiration) and low rainfall of high intensity and short duration [8–11]. Likewise, the infiltration rate of rainfall that feeds the aquifers is conditioned to the geological structure within the basin. Conceptual understanding of the geological framework in the recharge process is complex. Accurate quantitative assessments of multiple parameters are needed to find associations and identify potential groundwater recharge zones. Researchers have included many variables and classified them as determining the source of recharge (rainfall, drainage, irrigation, etc.) [12–17] and those that influence infiltration (soil, land use, slope, geology, lithology, hydraulic conductivity, etc.) [18–25].

Multivariate statistical approaches have been robust tools for managing groundwater resources [26,27]. These methods have been successfully applied in various disciplines [25]. Previous studies have used cluster analysis (CA) and principal component analysis (PCA) to identify groundwater pollution sources, assess water quality, analyze groundwater recharge processes, and conduct environmental studies [25–30].

To our knowledge, there are few studies that have applied *K-means* clustering algorithm and PCA on variables as soil, slope, geology, vegetation, and rainfall, to identify the dominant factors controlling the groundwater recharge [25,28]. This could be due to the fact that in most regions there is not a suitable spatio-temporal characterization of geophysical variables. These techniques have been applied chiefly in hydrogeochemical investigations [31–34]. Moreover, geospatial analysis techniques have been helpful to understand the characteristics of recharge [8,35].

The Sierra de San Miguelito Volcanic Complex (SSMVC) is located in a semi-arid environment in the San Luis Potosi Valley (SLPV). Previous research reports that the SSMVC is a recharge zone that feeds the deep aquifer of the SLPV [24,36,37]. Other studies mention that the SSMVC could not be working optimally as a recharge area [38,39]. Nowadays, it is uncertain whether the SSMVC functions as a groundwater recharge zone. Therefore, a methodology is proposed by applying and analyzing with *K-means* clustering algorithm and PCA, combined with geospatial analysis to identify main variables that determine groundwater recharge processes and define potential recharge zones in the SSMVC.

## 2. Materials and Methods

### 2.1. Description of the Study Area

The SSMVC is located in central Mexico, towards the western edge of the estate of San Luis Potosi; the study zone has an approximate extension of 149,669 ha (Figure 1). The area is steep and irregular, and it is characterized by having slopes greater than 30° and an altitude between 1900 and 2870 m above sea level (masl), classifying it as a high mountain range with plateaus [40–42]. The climate is semi-arid temperate, with an annual temperature that ranges between 20 and 22 °C [42]. Annual rainfall ranges between 400 and 600 mm, while evapotranspiration (2551 mm) exceeds average rainfall (408 mm) [42–44]. The predominant native plant species are *Pinus cembroides* Zucc. (1832) and *Quercus potosina* Trel. (1924), developed in soils classified as lithic-paralithic Leptosols in the highest parts of the sierra [41,45]. Other types of vegetation develop in the lower parts of the sierra, such as grasslands, chaparrals, and scrublands on soils of more significant proportion, such as regosols, phaeozems, and planosols [46].

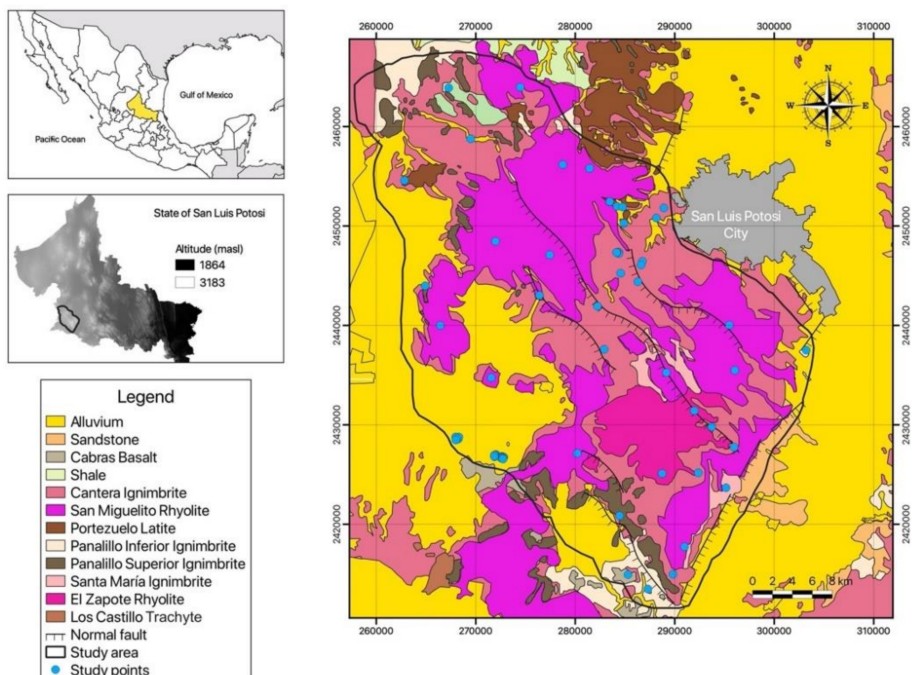

**Figure 1.** Location map of the study area, showing the study point's location and geology aspects.

### 2.1.1. Geology Settings

The geology of the SSMVC is of volcanic origin, made up of many silicic rocks from the Oligocene and Miocene (31 to 26 million years), where the Oligocene sequence is formed by lava flows of rhyolitic composition and ignimbrites. In contrast, the Miocene sequence is characterized by the emission of lavas that vary from basaltic (Cabras Basalt) to trachytes (Los Castillo Trachyte), this being the last volcanic activity in the area [47,48] (Figure 1). Structurally, the sierra contains several normal faults with strikes from 300° to 340°, and almost all have SW dip-directions ranging from 45° to 75° [49,50]. These faults were classified as a domino system because they show uniform fault dip direction and similar bed dip angles [50].

### 2.1.2. Hydrogeology Settings

The SLPV is limited to the west by the SSMVC and the east by the Sierra de Álvarez (SA); in the valley, the existence of two aquifers has been identified: (1) a shallow aquifer (hook) and (2) a deep aquifer (fractured volcanic), which are separated by a layer composed of compact fine sand with low hydraulic conductivity [16,51].

The material that makes up the shallow aquifer presents textural variations towards the SSMVC, conglomerates immersed in a clayey matrix predominate, and towards the SA this material thins, with silts and sands predominating [52,53]. The shallow aquifer has an approximate thickness of 5 to 40 m and a low hydraulic conductivity of $\approx 2 \times 10^{-4}$ m/s; due to its shallow depth, it is susceptible to seasonal effects [52,54].

Additionally, the deep aquifer is formed by fractured volcanic rock and is confined in the center of the SLPV by a low permeable sedimentary layer, it is limited on one side by the Sierra de San Miguelito and on the other side by the SA, and it has a low hydraulic conductivity of the order from $\approx 2 \times 10^{-4}$ m/s [43,46]. Its upper limit is approximately 100 to 150 m deep [39].

Between the shallow and deep aquifer is a granular layer, which is mainly composed of Quaternary clastic materials, the thickness of which varies from 100 to 200 m. This layer is confined in the center and is exploited by pumping wells with a depth of up to 350 m [52,55].

### 2.2. Research Approach

The approach used in this research included three components: Phase I Literature review, where the variables were selected and the data obtained from several sources; Phase II Raw data treatment, which included the generation of thematic maps and the study point selection; and Phase III, Multivariate statistical analysis, including PCA analysis and *K-means* clustering (Figure 2).

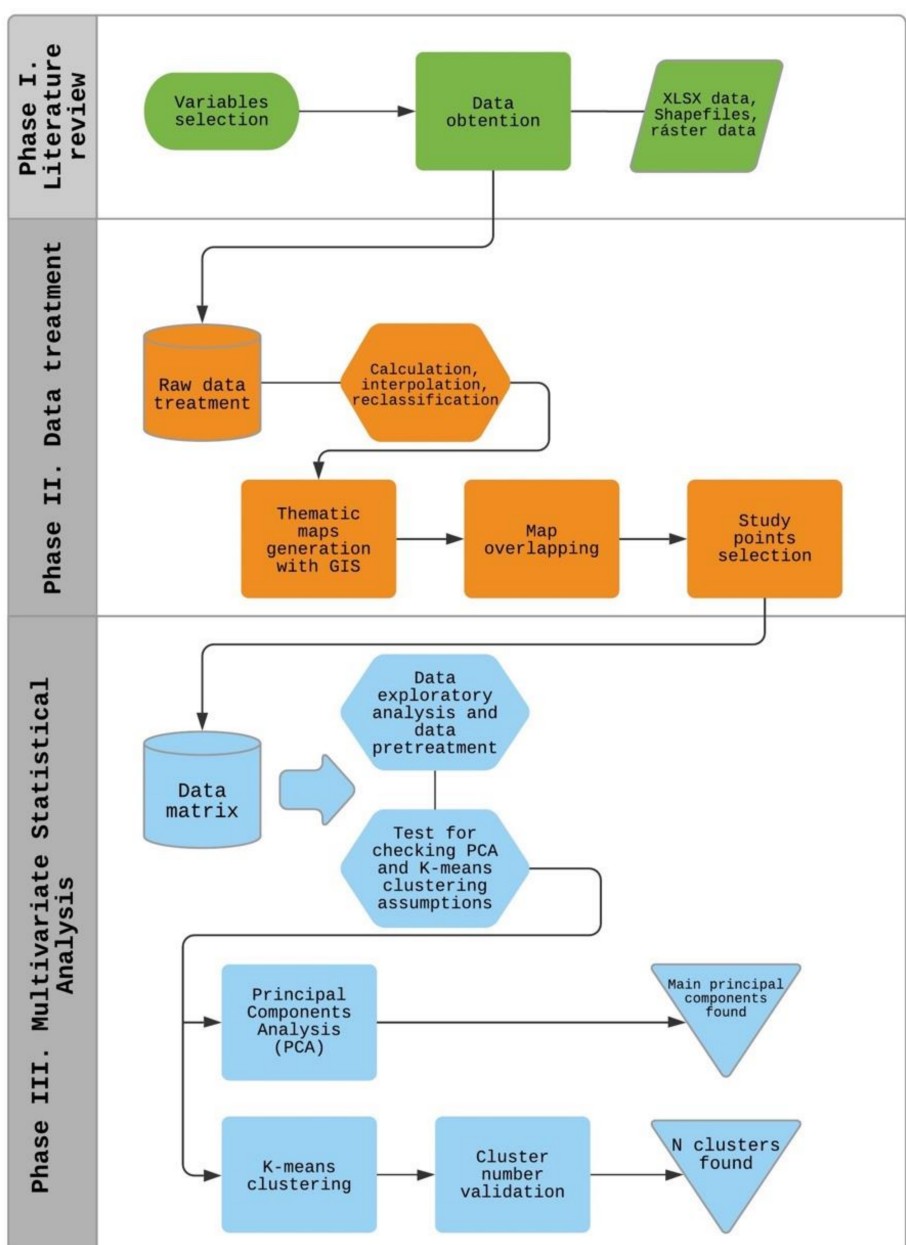

**Figure 2.** Processes followed for the description of potential groundwater recharge areas.

### 2.2.1. Phase I Literature Review

Twelve variables were selected: altitude (masl), slope (%), temperature (°C), soil type, vegetation type, rainfall (mm/year), relative humidity (%), potential evapotranspiration (PET, mm/year), land use, runoff coefficient, hydraulic conductivity (K, m/d), and geology. The data were obtained from INEGI and Climate Forecast System Reanalysis (CFSR) in shp, raster, and xlsx formats.

### 2.2.2. Phase II Data Treatment

To realize the corresponding thematic maps, vector data in shapefile format of land use and vegetation type and soil type were obtained from INEGI for the years 2013 and 2007, respectively. A geological map of the study area was made, taking reference from the maps used by [43,48].

The altitude and slope variables were obtained from the Mexican Continuous of Elevations (CEM 3.0 in Spanish) of INEGI for San Luis Potosi, extracting the level curves of the study zone and obtaining slope ranges, generating the respective thematic maps.

The thematic maps generated were made with QGIS v. 3.4.0 Madeira, which allows handling and analyzing spatial information as well as superimposing thematic layers. Cartographic information was used under the *WGS84* datum coordinate system at a scale of 1:250,000. Thematic layers of land use and vegetation, soil type, altitude, geology, and slope were overlapped to define the study points (Figure 3). Through the map overlap, attributes of each entity were intersected, allowing the identification of points with different characteristics as well as their spatial distribution. A total of 66 points were selected considering this criterion (Figure 1).

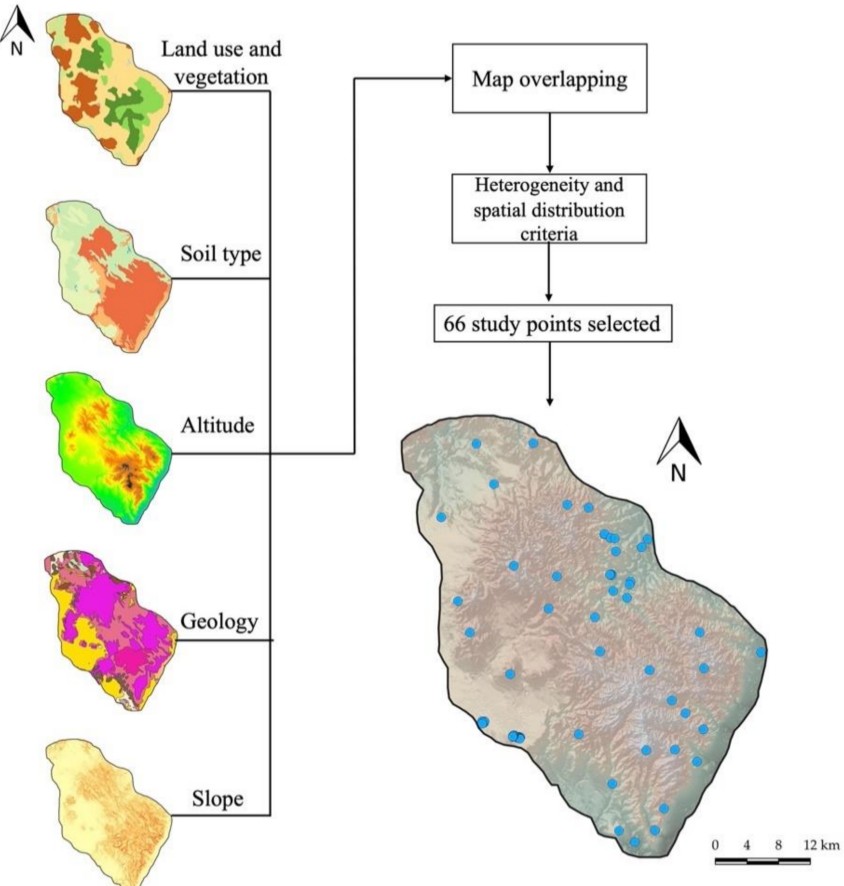

**Figure 3.** Scheme of the map overlapping procedure for the study points selection.

Data of rainfall, relative humidity, and temperature were employed. The data sets were downloaded from the CFSR for nine reference points close to the study area over a 34-year period (1979 to 2013). PET was calculated through Thornthwaite method [56] based on the mean temperature and astronomical duration of the day for a specific latitude (See Supplementary Materials, Table S1). An interpolation was carried out with the Inverse Distance Weighting (IDW) method, considering the average values obtained at each reference point. The values of climatic variables were obtained for each study point.

The runoff coefficient was determined, considering the Prevert Coefficient [57] (See Supplementary Materials, Table S2), which encompasses three conditions: land use, soil texture, and terrain slope. This coefficient was supplemented with the Chow Coefficient [58] (See Supplementary Materials, Table S3) because Prevert does not consider land use for urban areas. For the determination of K, data from previous studies in the SSMVC were taken [59,60], as well as values reported in hydraulic conductivity tests carried out in similar geological units and considering characteristics such as slope, altitude, proximity to the fault zone and fractures, etc. [61–67]. The values employed were transformed into meters per day (m/d). The runoff coefficient and K values were obtained for each study point.

### 2.2.3. Phase III Multivariate Statistical Analysis

A data matrix with 66 study points and 12 variables was integrated. Those qualitative variables were transformed into quantitative variables through coding to facilitate their handling.

The optimal results in applying multivariate statistical techniques require a univariate and multivariate normal distribution and homogeneity of variances (homoscedasticity) [26,27,68–70]. The univariate and multivariate normal distribution was verified by Shapiro-Wilk's test [71] and Royston's test [72], respectively. A non-normal distribution was observed on the data matrix in both tests. A logarithmic transformation (natural logarithm) was applied to the original data matrix to achieve a normal-like distribution. The Kolmogorov-Smirnov (K-S) test evaluated the adjustment of the transformed variables to the normal log distribution, being favorable ($p$-value < 0.05).

The database was scaled through standardization to achieve the optimal conditions for multivariate analysis. Standardization reduces the difference in variances in variables and prevents dissimilarity measures like the Euclidian distance obtained from being affected [27,73]. Each variable was standardized to their corresponding $Z$ scores, which were calculated by Equation (1):

$$Z_i = \frac{(X_i - mean)}{S} \tag{1}$$

where $Z_i$ is the standardized $Z$ score, $X_i$ is the value for each variable, and *mean* and $S$ are the mean value and the standard deviation of each variable, respectively.

The Kaiser Meyer Olkin (KMO) and Bartlett's sphericity tests were applied to assess the precision and suitability of the data for PCA. KMO is used to measure the adequacy of the sampling, designating the portion of shared variance. A value close to 1 commonly indicates that PCA may be useful [26,27,74]. In this study, a valid KMO value of 0.77 was obtained (See Supplementary Materials, Table S4). Bartlett's sphericity test allows to validate that the analyzed variables are adequately correlated; small values ($p$-value < 0.05), as were obtained in this study, indicate a great relationship between variables.

Correlation analysis is used to measure and establish the interrelation between two variables [75]. Based on the results obtained in the Shapiro-Wilk test, the most appropriate correlation method was selected [75,76]. Due to the variables presented a non-normal distribution ($p$-value < 0.05) (See Supplementary Materials, Table S5), the Spearman correlation method was employed. In this work, the criteria established by [27] were used; a value of $r$ greater than 0.7 indicates a high correlation, and a value of $r$ between [0.5, 0.7] denotes a moderate association.

Cluster analysis aims to classify a sample into small groups based on similarities between units and differences between groups [26,73,77]. The *K-means* algorithm was used, which divides the data into a number of clusters specified by the user and characterized by centroids [73,78]. In each repetition, the occurrences are assigned to the closest groups based on the Euclidean distance between instances and centroids (Equation (2)), such that

the squared error between the empirical means of a group and the points in the groups is minimized [77,78].

$$d(Z_p, Z_q) = \|Z_p - Z_q\|2 = \sqrt{\sum_{\substack{i \\ j=1}}^{D} (Z_{pj} - Z_{qj})} \qquad (2)$$

where $d$ is the distance, $Zp$ is the point in the space representing a given object, $Zq$ is cluster q, $Zpj$ is known as the $j$th attribute of the $p$th instance, $Zqj$ is the $j$th attribute of the $q$th cluster, and $D$ is known as the total number of attributes. The *K-means* grouping was formulated as the sum of the squared errors [26,77,79], as shown in Equation (3):

$$K = \sum_{l=1}^{k} \sum_{x \in C_l} \|x - m_l\|^2 \qquad (3)$$

where $X = \{x_1, ....., x_n\}$ is the data, $m_l = \sum x \in C_l \frac{x}{n_1}$ is known as the centroid of cluster $C_l$, $1 \le 1 \le K$, $n_1$ is the number of data objects in the cluster, and K is the number of clusters.

The main purpose of the PCA is to explain the variance within a data set while reducing the dimensionality of its structure [25,26,80]. PCA was carried out to transform the original correlated variables into a smaller set of uncorrelated variables called the principal components (PCs) [69,81,82]. PCs are expressed as loadings, which indicate the relative contribution of a given variable to each of the extracted PCs [81]. The principal component (PC) is expressed by Equation (4) according to [83]:

$$Z_{ij} = a_{i1}x_{1j} + a_{i2}x_{2j} + a_{i3}x_{3j} + \cdots + a_{im}x_{mj} \qquad (4)$$

where $Z$ is the component score, $a$ is the component loading, $x$ is the estimated value of the variable, $i$ is the component number, $j$ is the sample number, and $m$ is the total number of variables.

The PCs were chosen based on the eigenvalues (>1) and the cumulative percentage of the dataset variance.

The software RStudio v. 1.3.959 (Copyright RStudio Inc., Boston, MA, USA) was employed to perform data pretreatment, descriptive statistics, correlation matrix, *K-means* clustering algorithm, and PCA.

## 3. Results and Discussion

### 3.1. Correlation among Variables

In this study, Spearman's rank correlation (r) was applied to measure and determine the inter-variables relationships. The results obtained from Spearman's correlation analyses are shown in Table 1.

**Table 1.** Spearman's rank correlation matrix for the variables studied.

| Variables | Altitude | Slope | Temp | Soil | Vegetation | Rainfall | Relative Humidity | PET | Land Use | Runoff Coefficient | K | Geology |
|---|---|---|---|---|---|---|---|---|---|---|---|---|
| Altitude | 1 | | | | | | | | | | | |
| Slope | 0.12 | 1 | | | | | | | | | | |
| Temp | 0.05 | −0.34 | 1 | | | | | | | | | |
| Soil | 0.16 | −0.2 | 0.48 | 1 | | | | | | | | |
| Vegetation | −0.15 | −0.06 | 0.07 | 0 | 1 | | | | | | | |
| Rainfall | −0.12 | 0.28 | −0.75 | −0.12 | 0.07 | 1 | | | | | | |
| Relative humidity | 0.06 | 0.32 | −0.89 | −0.45 | −0.11 | 0.66 | 1 | | | | | |
| PET | 0.03 | −0.38 | 0.93 | 0.39 | 0.14 | −0.82 | −0.88 | 1 | | | | |
| Land use | 0.01 | 0.43 | −0.82 | −0.61 | −0.06 | 0.52 | 0.86 | −0.78 | 1 | | | |
| Runoff coefficient | 0.3 | 0.62 | −0.11 | 0.31 | −0.29 | 0.27 | 0.09 | −0.27 | 0.05 | 1 | | |
| K | 0.48 | 0.27 | −0.3 | −0.05 | −0.09 | 0.22 | 0.4 | −0.31 | 0.32 | 0.32 | 1 | |
| Geology | −0.44 | −0.53 | 0.52 | 0.45 | 0.16 | −0.18 | −0.59 | 0.53 | −0.69 | −0.30 | −0.41 | 1 |

Note: Coefficients greater than 0.5 and minor than −0.5 are underlined.

Temperature and PET have the highest correlation coefficient (0.93). This could be due to increase air temperature during the last years in the study zone [84]. Previous research has reported similar correlations between temperature and evapotranspiration [85,86]. This could be attributed to high temperatures increasing evaporative demand, inducing humidity deficits and evaporative stress which will result in an increase in evapotranspiration that is partly dependent on climate [87]. This could be explained by intensive irrigation practices and urban sprawl in the study zone. However, temperature showed a high inverse correlation with relative humidity (−0.89), suggesting that with the increase of temperature, the humidity decreases and vice versa at high humidity, the temperature declines [88,89]. Studies reported by [52] have demonstrated that a low level of humidity is related to high temperature levels in the SSMVC. Similar studies have documented that the changes in climatic conditions, such as an increase in temperature, are associated with greater soil humidity deficits. This can decrease the magnitude of groundwater recharge affecting water availability [90].

A high positive correlation between relative humidity and land use (0.86) was obtained; this could be explained by the agricultural irrigation practices that have been intensified, including large areas of land [91–93]. PET obtained a high inverse correlation of −0.88 and −0.82 with humidity and rainfall, respectively; this reveals that the humidity and precipitation decrease when the PET increase and vice versa. This can be due to the evapotranspiration exceeding the average annual rainfall [51–53]. In semi-arid regions, approximately 90% of the rainfall is lost through evapotranspiration [94]. Additionally, the impact of climatic characteristics on groundwater recharge is indirectly determined by the PET on soil humidity. For example, a drier soil could delay groundwater recharge and make it difficult [95]. In previous studies in other countries, researchers reported that high evapotranspiration rates were principally related to decreases in humidity [96]. A negative association (−0.75) between temperature and rainfall can be attributed to elevated temperatures (40 °C) in the SLPV, causing a rise in evaporation and, consequently, a decrease in rainfall [44,52].

An inverse association of −0.82 and −0.78 was observed by land use with temperature and PET, respectively; this suggests that land use changes have altered the spatio-temporal temperature patterns [97,98]. It has likewise been observed that the evapotranspiration increased. Previous studies have attributed the land use and land cover change to increased evapotranspiration. This is due to anthropogenic activities and climate change. Some researchers have reported that land use and land cover change have greater effect on the hydrological cycle and, consequently, groundwater recharge dynamics [95,96,99]. Moreover, land use is negatively correlated with geology (−0.69) and soil type (−0.61); this could be because geology plays a significant role in the current land use. The local geology defines the soil type and processes in soils, and has an important impact on chemical and hydraulic characteristics [100]. The movement of water is determined by soil texture and composition. For example, sandier soils tend to have high groundwater recharge rates, while clayey soils tend to have restricted water movement [101]. Therefore, water infiltration into the soil is influenced by soil hydraulic properties, precipitation rate, and the initial water content of the soil [102]. This could provide vital information on potential groundwater recharge areas [103].

A moderate positive correlation between rainfall and humidity (0.66) was observed; low levels of humidity belonging to low rainfall (350 to 400 mm/year) in the study area could explain it [52]. Previous investigations in other regions obtained similar associations of 0.46 and 0.59 between rainfall and humidity [104,105].

Slope and runoff coefficients showed a moderate positive correlation (0.62); this could be due to the irregular steep slope, which favors greater runoff in the study zone, indicating that the runoff increase would decrease infiltration and vice versa [106–108]. This could be explained by steep slopes tending to decrease groundwater recharge due to the runoff flowing rapidly. Meanwhile, plains tend to improve groundwater recharge, because higher retention time contributes to rainwater infiltration of soils [109].

Moderate inverse associations of −0.59 and −0.53 were observed by geology with humidity and slope, respectively; this could be related to the presence of potential groundwater recharge zones. The SSMVC is characterized by faults and fractures, which could favor water infiltration [110]. Similar studies have considered largely geologic features to identify recharge potential zones [111]. This is due to the recharge involving complex interplay factors during the infiltration process from vadose zone and saturate zone.

### 3.2. Cluster Analysis

Cluster analysis by the *K-means* clustering algorithm was applied to identify and define groups of sampling points based on their similarities and differences. The results obtained by the spatial cluster analysis are shown in Figure 4. Three principal groups were found with different characteristics related to their potential groundwater recharge.

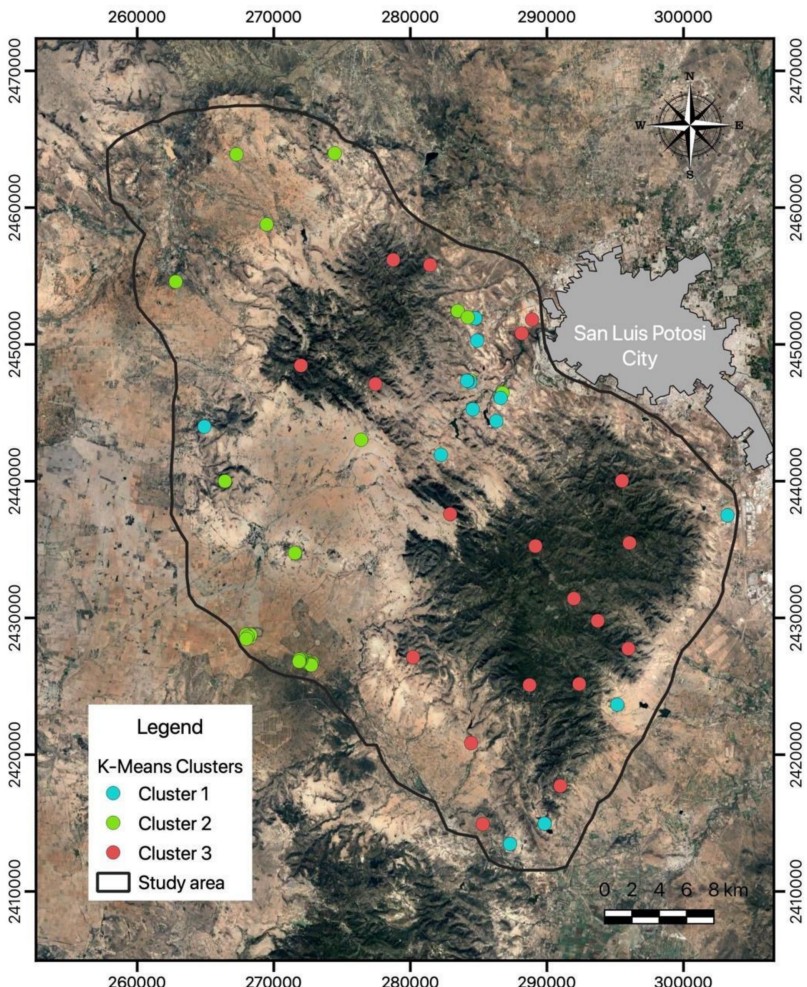

**Figure 4.** Spatial distribution map of groups using *K-means* clustering.

### 3.2.1. Cluster 1

The first class (Cluster 1) is located in northeastern and southeastern parts of the SSMVC. Cluster 1 includes 23% of the total points analyzed (Figure 4) and showed low potential for groundwater recharge. The first group showed gentle slopes (0–5%) in most of the land (Table 2); this can be linked to lower runoff coefficients (0.20) observed in Cluster 1 (Table 3). Similar studies have reported that gentle slopes would have more contact time between water and soil layer, decreasing runoff and increasing surface infiltration [106–108,112]. Moreover, surface crusts favor the redistribution of water and its accumulation, which supports vegetation conservation [113]. This can contribute to groundwater recharge in the study zone. However, the occurrence of recharge is limited by

the structure geologic, where group 1 was defined by ignimbrites and volcanic rocks of low porosity and hydraulic conductivity (Table 2). However, high hydraulic conductivity values (0.297 m/d) can be related to fractures or faults zones located in the southeastern and central parts, which could increase the hydraulic conductivity [61,63,114]. Therefore, diversities in the hydraulic properties and composition of superficial deposits could affect the subsurface dynamics of recharge to deeper aquifers [115–117]. Additionally, it found that natural grassland is the principal dominant vegetation of cluster 1 (Table 2). Previous researches have associated natural grassland and altitudes higher than 2300 masl to potential groundwater recharge [24]. Cluster 1 was characterized by an average altitude of less than 2300 masl (Table 3). This explains its low potential for groundwater recharge. The results are supported by similar studies, showing that the steep slopes with shallow rooted grasslands decrease surface runoff and favor infiltration, thus leading to an increase in groundwater recharge rates [118,119].

**Table 2.** Descriptive statistics for the groups found from *K-means* clustering analysis for the categorical variables.

| Categorical Variables | | | | |
|---|---|---|---|---|
| **Variable** | **Class** | **1 (*n* = 15)** | **2 (*n* = 32)** | **3 (*n* = 19)** |
| Slope | 0–5 | 86.67 | 84.38 | 36.84 |
| | 5–10 | 13.33 | 15.63 | 36.84 |
| | 10–30 | 0 | 0 | 21.05 |
| | >30 | 0 | 0 | 5.26 |
| Soil | Regosol | 73.33 | 28.13 | 10.53 |
| | Phaeozem | 20 | 0 | 78.95 |
| | Planosol | 0 | 65.63 | 0 |
| | Leptosol | 0 | 3.13 | 10.53 |
| | Fluvisol | 6.67 | 0 | 0 |
| Vegetation | Crassicaule shrubland | 6.67 | 9.38 | 5.26 |
| | Chaparral | 0 | 0 | 42.11 |
| | Oak | 0 | 0 | 10.53 |
| | Natural grassland | 93.33 | 90.63 | 21.05 |
| | Pine | 0 | 0 | 21.05 |
| Land use | Bare ground | 100 | 21.88 | 100 |
| | Temporary farming | 0 | 78.13 | 0 |
| Geology | Rhyolite | 0 | 12.50 | 57.89 |
| | Ignimbrite | 100 | 21.88 | 36.84 |
| | Basalt | 0 | 0 | 5.26 |
| | Alluvium | 0 | 62.50 | 0 |
| | Lutite | 0 | 3.13 | 0 |

Note: Data are represented in percentage (%).

**Table 3.** Descriptive statistics for the groups found from *K-means* clustering analysis for the numerical variables.

| Variable | Unit | 1 (*n* = 15) | | | | 2 (*n* = 32) | | | | 3 (*n* = 19) | | | |
|---|---|---|---|---|---|---|---|---|---|---|---|---|---|
| | | Min | Max | Mean | SD | Min | Max | Mean | SD | Min | Max | Mean | SD |
| Altitude | masl | 1900 | 2200 | 2046.67 | 74.32 | 2000 | 2300 | 2100 | 71.84 | 1900 | 2800 | 2368.42 | 260.45 |
| Temp | °C | 13.30 | 13.41 | 13.60 | 0.07 | 13.40 | 13.70 | 13.58 | 0.07 | 13.30 | 13.60 | 13.42 | 0.09 |
| Rainfall | mm | 500 | 520 | 513.33 | 6.17 | 440 | 520 | 501.25 | 18.27 | 490 | 520 | 514.21 | 9.61 |
| Relative humidity | % | 58 | 59 | 58.3 | 0.46 | 56 | 58 | 56.41 | 0.80 | 57 | 59 | 58.21 | 0.79 |
| PET | mm | 668 | 677 | 671.27 | 2.01 | 671 | 678 | 675.47 | 20.06 | 668 | 675 | 670.47 | 2.20 |
| Runoff coefficient | - | 0.15 | 0.4 | 0.20 | 0.09 | 0.15 | 0.6 | 0.31 | 0.09 | 0.30 | 0.88 | 0.44 | 0.15 |
| K | m/d | 0.005 | 0.864 | 0.297 | 0.37 | $6.39 \times 10^{-9}$ | 0.54 | 0.07 | 0.13 | 0.004 | 0.864 | 0.44 | 0.34 |

Min = minimum, Max = maximum, SD = indicates standard deviation, *n* = indicates number of analyzed points by cluster.

### 3.2.2. Cluster 2

The second class (Cluster 2) is located southwest and northwest of the SSMVC. Cluster 2 comprises 48% of the total number of points studied (Figure 4) and showed a high potential for groundwater recharge. The geological setting of Cluster 2 is mostly of alluvium (Table 2), which tends to increase hydraulic conductivity [28,67]. Groundwater recharge is significantly influenced by lithology based on natural topography, slope, faults, fracture extension, interbedded strata type, and their sequence stratigraphic in which highly compacted layers reduce the recharge [120]. Gentle slopes of 0–5% and low runoff coefficients were identified (Tables 2 and 3); this could increase of infiltration and the presence of groundwater recharge areas. However, the lowest values (mean = 0.07 m/d) of hydraulic conductivity were observed. Meanwhile, mean values of K of 0.29 m/d and 0.44 m/d were found in Clusters 1 and 3, respectively (Table 3); this may be due to the predominant soil types being planosols, soils with low hydraulic conductivity [46,121]. When soils are flooded during the rainy season, the water is employed in irrigation practices (Table 2). Although studies have reported that agricultural zones have high groundwater recharge potential [28,122], they are also associated with climatic patterns such as temperature, humidity, precipitation, and evapotranspiration. Therefore, high evapotranspiration rates and planosol soil type could limit the superficial infiltration, hindering the natural recharge towards the deep aquifer. In semi-arid environments, the occurrence of efficient recharge is determined by several soil properties, like the porosity, drainage patterns, slope, class of soils, and weather. Therefore, it is essential to understand the groundwater resource condition, their occurrence, movement, and surface–groundwater interactions in drylands [115,123].

### 3.2.3. Cluster 3

The third class (Cluster 3) is situated south and northeast of the study zone. Cluster 3 includes 29% of the total number of points sampled (Figure 4) and showed low potential for groundwater recharge. The highest altitude (2368 masl) was observed in Cluster 3, whereas lower altitudes of 2046 and 2100 masl were observed in Clusters 1 and 2, respectively (Table 3). Cluster 3 is characterized by high slopes between 10 and 30 % and slopes greater than 30%, while lower slopes were determined in Cluster 1 and Cluster 2 (Table 2). Previous investigations disclosed that steep slopes increase the surface runoff, hindering the contact between water and soil surface [106–108]; this can be linked to the highest runoff coefficient (mean = 0.44) obtained in Cluster 3, while lower values of 0.20 and 0.31 were found in Clusters 1 and 2, respectively (Table 3). The geological structure is mostly formed of rhyolite and ignimbrite (Table 2). These formations are made up of interbedded layers of volcanic rocks of low porosity and hydraulic conductivity. However, the highest hydraulic conductivity (mean = 0.44 m/d) was identified, whereas lower K of 0.29 and 0.07 m/d was observed in group 1 and group 2 (Table 3). This can be linked to fractures or faults in the study area, which increases the K. Studies have documented the variations in hydraulic properties influencing overall aquifer dynamic, defining referential flow directions, and modifying their storage capacity [61,63,65].

Three different vegetation types (chaparral and oak-pine forest) were identified in Cluster 3 (Table 2) and are associates with groundwater recharge [24]. Previous studies in the SSMVC have reported that the surface infiltration reaches a depth of 50 cm. This is explained by the interactions between pine forest and volcanic rock [41,43]. Therefore, the surface infiltrations rates are limited by the vegetation type and rock type and hinder the natural recharge. Studies reported that deep rooted ecosystem, such as the eucalypt forests, could decrease groundwater recharge. Meanwhile, steep slopes such as such as the ones in this group with shallow rooted grasslands reduce surface runoff and promote infiltration [118,119].

### 3.3. Principal Component Analysis

PCA was applied to reduce the high-dimensional dataset to a small dataset with most of the information of the initial dataset. PCA was performed on a dataset consisting of 66 observations and 12 variables. Four components (PCs) were defined, accounting for 77.88% of the total variance in the dataset. The results are shown in Table 4 and Figure 5.

**Table 4.** Summary of the PCA loadings on the variables.

| Variables | Principal Component Matrix | | | |
|---|---|---|---|---|
| | PC1 | PC2 | PC3 | PC4 |
| Altitude | 0.328 | 0.444 | −0.259 | 0.644 |
| Slope | 0.520 | 0.570 | 0.140 | −0.278 |
| Temp | −0.909 | 0.185 | −0.136 | 0.104 |
| Soil | −0.491 | 0.395 | 0.440 | 0.306 |
| Vegetation | −0.107 | −0.415 | 0.588 | 0.280 |
| Rainfall | 0.568 | −0.156 | 0.704 | 0.127 |
| Relative humidity | 0.936 | −0.141 | −0.004 | 0.030 |
| PET | −0.928 | 0.073 | −0.102 | 0.103 |
| Land use | 0.916 | −0.181 | −0.099 | −0.057 |
| Runoff coefficient | 0.140 | 0.783 | 0.363 | −0.317 |
| K | 0.530 | 0.206 | −0.056 | 0.452 |
| Geology | −0.798 | −0.138 | 0.317 | −0.075 |
| Eigenvalue | 5.293 | 1.642 | 1.395 | 1.017 |
| Variability (%) | 44.106 | 13.683 | 11.626 | 8.471 |
| Cumulative (%) | 44.106 | 57.789 | 69.415 | 77.886 |

Note: Coefficients greater than 0.5 and minor than −0.5 are underlined.

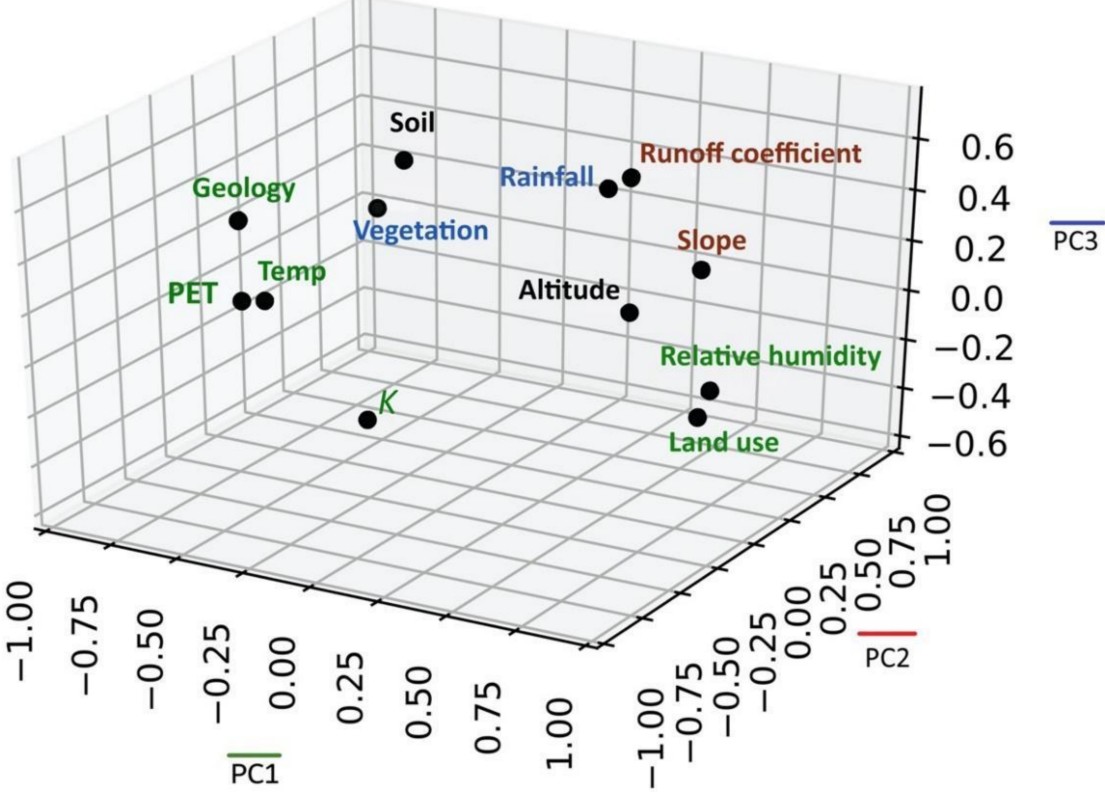

**Figure 5.** Principal Component Analysis (PCA) plot of the variables.

The first component explains 44.10% of the total variance. PC1 has a robust negative loading of −0.92, −0.90, and −0.79 on PET, temperature, and geology, and high positive loading of 0.93, 0.91, 0.56, 0.53, and 0.52 on humidity, land use, precipitation, hydraulic conductivity, and slope. PC1 suggests that the occurrence of infiltration processes is determined by hydraulic conductivity, rainfall, and humidity in the SSMVC. Potential groundwater recharge zones are controlled by geological formations, high levels of PET, and temperature. In previous studies, researchers have recognized water deficits that caused low infiltration rates [11,15]. This is warranted by the high levels of evapotranspiration and temperature observed in the SSMVC. Previous studies have reported that the recharge depends on the nature and hydraulic properties of the vadose zone. This is due to the fact that the unsaturated layers are often defined by the permeability and porosity features which vary from point to another. Thus, groundwater recharge is governed by subsurface geology. In addition, the occurrence of fractures and faults are linked to increase of water movement (high hydraulic conductivity) to saturated zone [111]. However, variation of natural recharge process associated with PET and temperature is due to the groundwater recharge being controlled, in part, by precipitation and evaporation, itself dependent on temperature [124,125]. A high association between humidity and land use can be due to a large area of agricultural fields in the study zone. Studies have demonstrated that the zones with high groundwater recharge potential are located in areas with dense vegetation cover coupled with flat relief and consolidated and structured soils [126].

The second component describes 13.68% of the variance of the dataset. PC2 has a fit, positive loading of 0.78 and 0.57 on runoff coefficient and slope. This indicates that the hillslopes have a significant function in the rainfall-runoff processes, determining the infiltration rates; thus, groundwater recharge [106–108]. It observed a suite of cluster analyses where gentle slopes in Cluster 1 and Cluster 2 were linked to lower runoff coefficients. Meanwhile, high slopes in Cluster 3 were related to the highest runoff coefficient. In previous studies, researchers have highlighted that suitable slope and elevation delineated by basin and depressions are crucial for recharge. This is due to the runoff being larger over zones of steeper slopes, leaving no time for infiltration. Thus, it is probable that recharge is decreased as runoff flows faster and does not allow infiltration [109,127,128].

The third component explains 11.62% of the total variance. PC3 has a robust positive loading of 0.70 and 0.58 on precipitation and vegetation. This reveals the strong influence of precipitation and vegetation on the increase in evapotranspiration. Previous studies have reported that the surface soil layers are recharged by rainfall and lose water by high evapotranspiration rates [39,53]. It observed that the infiltration rates were defined by high precipitation (514.21 mm) and vegetation type [41,43]. Previous investigations have described that landslides and floods in the study zone could be attributed to high runoffs [46]. It is well known that the temperature and precipitation can have meaningful effects on groundwater recharge [129,130]. Particularly the recharge is significant in semi-arid regions where rainfall is variable and evapotranspiration frequently exceeds precipitation [109,131]. In addition, previous studies have found a high association between recharge and both climate and land cover [125]. Some researchers have documented that variations in groundwater recharge are strongly related to vegetation types [129].

The fourth component explains 8.47% of the variance of the data. PC4 has a robust positive loading of 0.64 on altitude. Similar investigations in other countries have reported that the potential groundwater recharge zones are determined by low altitudes [132]. However, it observed that the altitudes of up to 2800 masl caused increased runoff and decreased soil infiltration rate. This indicates that the groundwater recharge is controlled by altitude, such as in Cluster 3, where the highest altitude and runoff coefficient was found.

## 4. Conclusions

The results obtained from correlation analysis indicated that climatic variables such as temperature, humidity, precipitation, and evapotranspiration determine the groundwater recharge process, while land use and geology define potential recharge zones. A high

correlation between temperature and evapotranspiration (0.93), humidity (−0.87), and rainfall (−0.75) was observed. This is explained by an increase in the temperature during the last years in the study zone, which has altered the spatio-temporal rainfall patterns. Moreover, land use showed a robust negative association with local geology (−0.69) and soil type (−0.61). This could be due to the fact that the local geology determines the soil type and soil processes and has a meaningful influence on the hydraulic conductivity, which provides primary information on potential recharge zones.

Statistical methods such as *K-means* clustering and PCA were usefully applied to identified main factors that determine recharge processes and potential groundwater recharge areas. The *K-means* algorithm recognized three clusters. The first group showed a low potential for groundwater recharge. This cluster was located in the southeastern and central parts. This group is characterized by gentle slopes (0–5%) in most of the land and lower runoff coefficients (0.20), as well as low porosity and hydraulic conductivity. However, it was possible to identify high hydraulic conductivity values (0.297 m/d), which can be related to fractures or faults zones. The second group disclosed a potential for groundwater recharge due to its geology and land use, but it is limited by h climatic factors. This cluster was located in the north and northwest portions. The geological setting of this group is mostly of alluvium, which tends to increase hydraulic conductivity. The third cluster revealed low potential for water recharge. This group is situated in the south and northeast parts. This cluster is characterized by the highest altitude (2368 masl), high slopes (>30%), which can be linked to the highest runoff coefficient (0.44) observed in Cluster 3. The geological structure of this cluster is mostly formed of rhyolite and ignimbrite of low porosity. However, it had the highest hydraulic conductivity (0.86), which can be related to fractures found in the study area.

Four components identified in the principal component analysis are responsible for 77.88% of the total variance in the data matrix, and were found to be the main variables controlling groundwater recharge: geology, K, temperature, precipitation, PET, humidity, and land use. Infiltration processes are restricted by low hydraulic conductivity, as well as ignimbrites and volcanic rocks of low porosity in the study area.

Given the climatic and geological conditions shown in this study, the SSMVC is not working optimally as a water recharge zone towards the deep aquifer of the SLPV. The methodology used in this study will be useful for water resource managers to develop strategies to define priority recharge areas and safeguard the sustainable management of water resources.

**Supplementary Materials:** The following are available online at https://www.mdpi.com/article/10.3390/su132011543/s1, Table S1: Thornthwaite method for the PET estimation, Table S2: Runoff coefficient of Prevert (1984), Table S3: Runoff coefficient of Chow (1993), Table S4: Results of the KMO test, Table S5: Results of the Shapiro-Wilk test.

**Author Contributions:** Conceptualization, A.E.M.C.; Data curation, J.L.U.C.; Formal analysis, J.L.U.C., D.A.M.C. and A.E.M.C.; Investigation, J.L.U.C.; Methodology, D.A.M.C. and A.E.M.C.; Resources, J.A.R.L.; Software, J.L.U.C. and A.E.M.C.; Supervision, A.E.M.C.; Validation, J.L.U.C. and A.E.M.C.; Visualization, J.A.R.L., D.A.M.C. and A.C.M.; Writing—original draft, J.L.U.C. and A.E.M.C.; Writing—review & editing, J.A.R.L., D.A.M.C., A.C.M. and A.E.M.C. All authors have read and agreed to the published version of the manuscript.

**Funding:** This research received no external funding.

**Institutional Review Board Statement:** Not applicable.

**Informed Consent Statement:** Not applicable.

**Data Availability Statement:** Not applicable.

**Acknowledgments:** J.L.U.C. thanks the Consejo Nacional de Ciencia y Tecnología (CONACYT) for a Ms.S. scholarship No. 930739 and the Instituto Potosino de Investigación Científica y Tecnológica A.C. (IPICYT).

**Conflicts of Interest:** The authors declare no conflict of interest.

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
