# Peer review of "Identification of the Dominant Factors in Groundwater Recharge Process, Using Multivariate Statistical Approaches in a Semi-Arid Region"

_sustainability, doi:10.3390/su132011543_

Round 1

Reviewer 1 Report

Manuscript ID: sustainability-1382369

Title: Identification of the dominant factors in groundwater recharge process, using Multivariate Statistical Approaches in a semi-arid region

This study identified the main variables that determine the recharge processes and definition of potential recharge zones on the San Luis Potosi Valley (SLPV). The data set of hydrological, morphological, geological, climatic and land use was used and analyzed using various statistical methods. The results are interesting.

Specific comments:

  • The scale in the Fig. 3 was not clear.
  • Please give the specific information on how to select the study points. The 66 points are not evenly distributed. You only mentioned “considering heterogeneous characteristics and spatial distribution”. Please give the specific information on how to select the study points.
  • You draw conclusion according to the results from K-means clustering and PCA. How did you verify you results? Can use monitoring data to prove?

Author Response

Comments and Suggestions

The changes requested are colored yellow

Answers

The changes made are colored green

The scale in the Fig. 3 was not clear.

Thanks a lot for your observation. We changed the size and clarity of the bar scale in the Figure 3.

Figure 3. Scheme of the map overlapping procedure for the study points selection

(Line: 162)

Please give the specific information on how to select the study points. The 66 points are not evenly distributed. You only mentioned “considering heterogeneous characteristics and spatial distribution”. Please give the specific information on how to select the study points.

Thank you very much for your observation. We made the changes and gave the specific information on how to select the study points. The changes are shown in the next paragraph:

Through the map overlap, attributes of each entity were intersected, allowing the identification of points with different characteristics as well as their spatial distribution. A total of 66 points were selected considering this criterion.

(Lines: 156-159)

You draw conclusion according to the results from K-means clustering and PCA. How did you verify you results? Can use monitoring data to prove?

Thank you very much for your comments.

It would be desirable to validate the results. However, in order to accomplish this, it is necessary to realize geophysical investigations using Electrical Resistivity Tomography with borehole data and field observations. The identification of geoelectrical structures associated with the lithology will allows recognize potential recharge pathways. The methodology used in this study can represent a previous phase of these studies, which have high costs. The geophysical studies could optimize the recharge zones previously identified with this methodology applied through of multivariate techniques and geostatistical analysis.

Reviewer 2 Report

This manuscript used multivariate statistical approaches to identify the dominant factors in groundwater recharge process in a semi-arid region. However, the value and scientific rigor are limited. Several specific comments are as follows.

  1. The abstract is too popular and not scientific enough to reflect the conclusion of the manuscript. Some sentences should be in the introduction.
  2. Lines 59-63 are somewhat paradoxical to read. Please rewrite them. Lines 64-65, Why “the study of groundwater recharge with K-means clustering algorithm and 64PCA on variables such as soil, slope, geology, vegetation, and rainfall, has been scarce”? Whether the research method is effective or not.
  3. Table 1 shows the spearman’s rank correlation matrix for the variables studied, but lacks the variation characteristics of these variables themselves. Therefore, the influence process and influence degree of these variables are not clearly clarified.
  4. In this study, statistical techniques, e.g. K-means cluster analysis and principal component analysis are used. One feature of K-means clustering is to check whether the classification of each sample is correct at the iteration. If it is incorrect, it will be adjusted to affect the accuracy of the experiment. Please pay attention to the sample classification to ensure its accuracy.
  5. Structuring up to level 4 (2.2.3.1., 2.2.3.2, etc.) should be avoided. This structuring makes the text harder to read.
  6. The part 3.3.1-3.3.4 don't look like four paragraphs. They should be contrast analysis in one paragraph.
  7. Discussion section is lack in this paper, which leads this paper to moderate quality. A profound discussion is strongly recommended.
  8. What is the accuracy when performing IDW on temperature and precipitation? Which is better compared to Kriging interpolation?
  9. In line 217“and a valuer of r between >0.5 o ≤0.7 denotes a moderate association.” Can it be represented by [0.5, 0.7], and “valuer” is mean “value”?
  10. There are many grammatical mistakes in the paper. Check them carefully.
  11. In line 23: “The Sierra de San Miguelito Volcanic Complex (SSMVC) belong to the San Luis Potosí Valley (SLPV)……”

In line 161: “PET was calculated through the Thornthwaite method based on the mean temperature and astronomical duration of the day for a specific latitude”

In line 212: “Correlation analysis allows to measure and establish the interrelation between two variables.”

In line 307: “Cluster 1 includes 23 % of the total number of points analyzed (Figure 4).”

In line 410: “PCA was applied to reduces the high-dimensional dataset to……”

Punctuation in line 216: “the criteria established by [27] were used,; a value of r greater than 0.7 indicates……”

Author Response

Comments and Suggestions

The changes requested are colored yellow

Answers

The changes made are colored green

The abstract is too popular and not scientific enough to reflect the conclusion of the manuscript. Some sentences should be in the introduction.

Thanks you very much for your observation and recommendation. We improved the abstract.

Identify potential groundwater recharge zones is essential for sustainable management of the resource in arid regions. An approach to define recharge areas is through of the identification of factors that govern their occurrence. Due to the fact that the recharge involves a complex interplay factors. In this study, a data matrix with 66 observations and climatic, hydrogeological, morphological, and land use variables was analyzed. The dominant factors in groundwater recharge process and potential recharge zones were appraised using K-means clustering, Principal Component Analysis (PCA) and geostatistical analysis. The study highlights the usefulness of use multivariate methods coupled with geospatial analysis to identify main factors that determine recharge processes and delineate potential groundwater recharge areas. Potential recharge zones were defined into cluster 1 and cluster 3, which were classified as low potential for recharge and cluster 2 was classified as high potential for groundwater recharge. Cluster 1 was located on flat land and nearby faults. It is mostly composed of ignimbrites and volcanic rocks of low hydraulic conductivity (K). Cluster 2 was ubicated in flat lowland agricultural, and it is mainly composed of alluvium, increasing hydraulic conductivity. Cluster 3 was located on steep slopes and closet to faults. It is formed of rhyolite and ignimbrite with interbedded layers of volcanic rocks of low hydraulic conductivity. PCA disclosed that groundwater recharge processes are controlled by geology, K, temperature, precipitation, PET, humidity, and land use. Infiltration processes are restricted by low hydraulic conductivity, as well as ignimbrites and volcanic rocks of low porosity. This methodology will be useful for water resources managers to identify and delineate potential recharge areas with greater certainty.

(Lines: 21-39)

Lines 59-63 are somewhat paradoxical to read. Please rewrite them. Lines 64-65, Why “the study of groundwater recharge with K-means clustering algorithm and 64PCA on variables such as soil, slope, geology, vegetation, and rainfall, has been scarce”? Whether the research method is effective or not.

Thanks a lot for your observation and comment. We have rewritten the paragraphs. The changes are shown in the next lines:

Multivariate statistical approaches have been robust tools for managing groundwater resources [26,27]. These methods have been successfully applied in various disciplines [25]. Previous studies have used Cluster Analysis (CA) and Principal Component Analysis (PCA) to identify groundwater pollution sources, assess water quality, analyze groundwater recharge processes, and environmental studies [25–30]

(Lines: 61-65)

To our knowledge, there are few studies that have applied K-means clustering algorithm and PCA on variables as soil, slope, geology, vegetation, and rainfall, to identify the dominant factors controlling the groundwater recharge [25,28]. This could be due to the fact in most regions there is no a suitable spatio-temporal characterization of geophysical variables.

(Lines:66-70)

Table 1 shows the spearman’s rank correlation matrix for the variables studied, but lacks the variation characteristics of these variables themselves. Therefore, the influence process and influence degree of these variables are not clearly clarified.

Thanks you very much for your observation. We improved the discussion. The changes are shown in the next paragraphs:

This could be attributed to high temperatures increase evaporative demand, inducing humidity deficits and evaporative stress which will result in increase in evapotranspiration that partly dependent on climate [87]. This could be explained by intensive irri-gation practices and urban sprawl in the study zone.

(Lines:252-255)

Additionally, the impact of climatic characteristics on groundwater recharge is indirectly determined by the PET on soil humidity. For example, in a drier soil could delay groundwater recharge and make it difficult [94]. In previous studies in other countries, researchers reported that high evapotranspiration rates were principally related to decreases in humidity [95].

(Lines:266-270)

Previous studies have attributed the land use and land cover change with increase evapotranspiration. This due to anthropogenic activities, and climate change. Some researchers have reported that land use and land cover change have greater effect on the hydrological cycle and, consequently, groundwater recharge dynamics [94,95,98].

(Lines:277-280)

The movement of water is determined by soil texture and composition. For example, sandier soils inclining to have high groundwater recharge rates, while those clayey soils tending to have restricted water movement [100]. Therefore, water infiltration into the soil is influenced by soil hydraulic properties, precipitation rate, and the initial water content of the soil [101].

(Lines:284-288)

This could be explained by steep slopes tending to decrease groundwater recharge due to the runoff flows rapidly. Meanwhile, plain tending to improve groundwater re-charge, because higher retention time is contributed for rainwater to infiltrate of soils [108].

(Lines:297-299)

In this study, statistical techniques, e.g. K-means cluster analysis and principal component analysis are used. One feature of K-means clustering is to check whether the classification of each sample is correct at the iteration. If it is incorrect, it will be adjusted to affect the accuracy of the experiment. Please pay attention to the sample classification to ensure its accuracy.

Thanks for your time in reviewing our article. In regards to your request:

The resulting outputs are highly sensitive to the initial numbers of cluster. For that reason, we paid attention to determine the optimal number of clusters and to the sample classification obtained to know its accuracy. Broadly speaking, this process was developed in three steps, which are described as follows

1.      We analyze the database to determine the initial numbers of cluster with the Nbclust package. The figure below shows that the optimal number of Cluster obtained (K=3)

2.      We performed a random partitioning of the database to obtain a training and test data set with the library ("XLConnect"). Next, we created a 3-group K-Means cluster model based on the training data, and used the "predict" function in the model calculated above to classify the test data.

3.       

We computed the accuracy of the obtained k-mean classification model. Using GIS, we label, as true, the test data according to the model based on the training data. We use the function "accuracy_score" which calculates: the set of predicted labels for a sample must exactly match the set of corresponding labels as true. If the set of predicted labels for a sample strictly matches the set of true labels, the subset accuracy is 1.0; otherwise, it is 0.0. Given the above parameters, we obtain a precision metric of 0.7.

Structuring up to level 4 (2.2.3.1., 2.2.3.2, etc.) should be avoided. This structuring makes the text harder to read.

Thank you very much for your observation. We removed the level 4 in the structure of the manuscript.

Due to the new structure, we relocated the next paragraph:

The software RStudio v. 1.3.959 (Copyright RStudio Inc., Boston, MA, USA) was employed to perform data pretreatment, descriptive statistics, correlation matrix, K-means clustering algorithm and PCA.

(Lines: 241-243)

Also, we removed the abbreviation “CA” for cluster analysis. The change is shown in the next line:

Cluster Analysis aims to classify a sample into small groups based on similarities…

(Line: 214)

The part 3.3.1-3.3.4 don't look like four paragraphs. They should be contrast analysis in one paragraph.

Thank you very much for your observation. We made the analysis in one paragraph. The changes are shown in the next paragraph:

The first component explains 44.10% of the total variance. PC1 has a robust negative loading of -0.92, -0.90 and 0.79 on PET, temperature and geology, and high positive loading of 0.93, 0.91, 0.56, 0.53 and 0.52 on humidity, land use, precipitation, hydraulic conductivity, and slope. PC1 suggests that the occurrence of infiltration processes is determined by hydraulic conductivity, rainfall, and humidity in the SSMVC. A high as-sociation between humidity and land use can be due to a large area of agricultural fields in the study zone. Potential groundwater recharge zones are controlled by geological formations, high levels of PET and temperature. In previous studies, researchers recognize water deficits that caused low infiltration rates [11,15]. This is warranted by the high levels of evapotranspiration and temperature observed in the SSMVC. The second component describes 13.68% of the variance of the dataset. PC2 has a fit, positive loading of 0.78 and 0.57 on runoff coefficient and slope. This indicates that the hillslopes have a significant function in the rainfall-runoff processes, determining the infiltration rates; thus, groundwater recharge [105–107]. It observed a suite of cluster analyses where gentle slopes in cluster 1 and cluster 2 were linked to lower runoff coefficients. Meanwhile, high slopes in cluster 3 were related to the highest runoff coefficient. The third component explains 11.62% of the total variance. PC3 has a robust positive loading of 0.70 and 0.58 on precipitation and vegetation. This reveals the strong influence of precipitation and vegetation on the increase in evapotranspiration. Previous studies have reported that the surface soil layers are recharged by rainfall and lose water by high evapotranspiration rates [39,53]. It observed that the infiltration rates were defined by high precipitation (514.21 mm) and vegetation type [41,43]. Previous investigations have described that landslides and floods in the study zone could be attributed to high runoffs [46]. The fourth component explains 8.47% of the variance of the data. PC4 has a robust positive loading of 0.64 on altitude. Similar investigations in other countries have reported that the potential groundwater recharge zones are determined by low altitudes [122]. However, it observed that the altitudes of up to 2800 masl caused increased runoff and decreased soil infiltration rate. This indicates that the groundwater recharge is controlled by altitude, such as in cluster 3, where the highest altitude and runoff coefficient was found.

(Lines: 447-469, 474-479)

Discussion section is lack in this paper, which leads this paper to moderate quality. A profound discussion is strongly recommended.

Thanks a lot for your recommendation. We made a profound discussion. The changes are shown in the next paragraphs.

This could be attributed to high temperatures increase evaporative demand, inducing humidity deficits and evaporative stress which will result in increase in evapotranspiration that partly dependent on climate [87]. This could be explained by intensive irrigation practices and urban sprawl in the study zone. 

(Lines: 252-255)

Additionally, the impact of climatic characteristics on groundwater recharge is indirectly determined by the PET on soil humidity. For example, in a drier soil could delay groundwater recharge and make it difficult [94]. In previous studies in other countries, researchers reported that high evapotranspiration rates were principally related to decreases in humidity [95].

(Lines: 266-270)

Previous studies have attributed the land use and land cover change with increase evapotranspiration. This due to anthropogenic activities, and climate change. Some researchers have reported that land use and land cover change have greater effect on the hydrological cycle and, consequently, groundwater recharge dynamics [94,95,98].

(Lines: 277-280)

The movement of water is determined by soil texture and composition. For example, sandier soils inclining to have high groundwater recharge rates, while those clayey soils tending to have restricted water movement [100]. Therefore, water infiltration into the soil is influenced by soil hydraulic properties, precipitation rate, and the initial water content of the soil [101].

(Lines: 284-288)

This could be explained by steep slopes tending to decrease groundwater recharge due to the runoff flows rapidly. Meanwhile, plain tending to improve groundwater re-charge, because higher retention time is contributed for rainwater to infiltrate of soils [108].

(Lines: 297-299)

Moreover, surface crusts favor to the redistribution of water and its accumulation, which support of vegetation conservation [111]. This can contribute to groundwater recharge in the study zone. 

(Lines: 326-328)

Therefore, diversities in the hydraulic properties and composition of superficial deposits could affect the subsurface dynamics of recharge to deeper aquifers [113-115]. 

(Lines: 333-335)

This explains its low potential for groundwater recharge. The results are supported by similar studies, showing that the steep slopes with shallow rooted grasslands decrease surface runoff and favors infiltration, thus leading to an increase in groundwater recharge rates [116-117].

(Lines: 338-341)

Groundwater recharge is significantly influenced by lithology based on natural topography, slope, faults, fracture extension, interbedded strata type and their sequence stratigraphic in which highly compacted layers reduce the recharge [118].

(Lines: 383-386)

In semi-arid environments, the occurrence of efficient recharge is determined by several soil properties, like the porosity, drainage patterns, slope, class of soils and weather. Therefore, it is essential to understand the groundwater resource condition, their occurrence, movement and surface-groundwater interactions in drylands [113-121]. 

(Lines: 397-401)

Studies have documented the variations in hydraulic properties influencing overall aquifer dynamic, defining referential flow directions and modifying their storage capacity [61,63,65] 

(Lines: 429-431)

Studies reported that deep rooted ecosystem such as the eucalypt forests, could de-crease groundwater recharge. Meanwhile, steep slopes such as such as the ones in this group with shallow rooted grasslands reduce surface runoff and promoting infiltration [116-117].

(Lines: 437-440)

What is the accuracy when performing IDW on temperature and precipitation? Which is better compared to Kriging interpolation?

Thanks a lot for your comments.

Interpolation models were constructed using IDW and Kriging algorithms. And the accuracy of the model rasters was evaluated. In the case of IDW interpolation, accuracy was evaluated by means of the RSME index obtained from the "metrics" package. El RMSE computes the average squared difference between two numeric vectors. In this way, the actual values were compared with those estimated by the interpolation model. An RMSE of 78 mm was obtained for precipitation and an RSM value of 0.22 ºC for temperature. Considering the ranges of these interpolated variables, we found the IDW models to be adequate compared to the Kriging models, which had larger biases. This analysis is shown in Figures 1 and 2.

a)      RMSE for IDW precipitation interpolation model

b)      RMSE for IDW temperature interpolation model

Figure 1. RMSE calculation of the models obtained by IDW interpolation.

In the case of Kriging interpolation, a variogram was constructed to evaluate the performance of this model. We build up a variogram model of spherical shape, which shows a poor fit of the theoretical model (green line) and the empirical model obtained blue dots, for both cases, precipitation and temperature.

It was needed to solve the linear equation system while assuring that the weights sum up to one. This factor can in turn be added to the weighted target semi-variances used to build the equation system, to obtain the Kriging error. If you interpolate these values you can also produce a map showing in which regions the interpolation is more certain.

a)      Variogram and Kriging error for precipitation

b)      Variogram and Kriging error for temperature

Figure 2. Interpolation and Kriging error map obtained by kriging interpolation.

In line 217“and a valuer of r between >0.5 o ≤0.7 denotes a moderate association.” Can it be represented by [0.5, 0.7], and “valuer” is mean “value”?

Thanks a lot for your observation. We represented adequately “moderate association” and corrected the word “valuer” as “value”. The change is shown in the next line:

….a high correlation, and a value of r between [0.5, 0.7] denotes a moderate association.

(Line: 213)

In line 23: “The Sierra de San Miguelito Volcanic Complex (SSMVC) belong to the San Luis Potosí Valley (SLPV)……”

Thank you very much for your observation. We corrected the grammatical mistake. The change is shown in the next line:

The Sierra de San Miguelito Volcanic Complex (SSMVC) is located……

(Line: 73)

In line 161: “PET was calculated through the Thornthwaite method based on the mean temperature and astronomical duration of the day for a specific latitude”

Thank you very much for your observation. We corrected the grammatical mistake. The change is shown in the next lines:

PET was calculated through Thornthwaite method [56], based on the mean temperature and astronomical duration of the day for a specific latitude.

(Lines: 165-167)

In line 212: “Correlation analysis allows to measure and establish the interrelation between two variables.”

Thank you very much for your observation. We corrected the grammatical mistake. The change is shown in the next line:

Correlation analysis is used to measure and establish the interrelation between two…

(Line: 208-209)

In line 307: “Cluster 1 includes 23 % of the total number of points analyzed (Figure 4).”

Thank you very much for your observation. We corrected the grammatical mistake. The change is shown in the next line:

Cluster 1 includes 23% of the total points analyzed (Figure 4) ...

(Line: 321)

In line 410: “PCA was applied to reduces the high-dimensional dataset to……”

Thank you very much for your observation. We corrected the grammatical mistake. The change is shown in the next line:

PCA was applied to reduce the high-dimensional dataset to a small dataset with most…

(Line: 443)

Punctuation in line 216: “the criteria established by [27] were used,; a value of r greater than 0.7 indicates……”

Thank you very much for your observation. We corrected the punctuation. The change is shown in the next line:

the criteria established by [27] were used; a value of r greater than 0.7 indicates…

(Line: 212)

Reviewer 3 Report

A statistical approach to hydrogeological data has been pretty much common last years. Thus the presented manuscript is not an innovative one but gives a broad outlook on the part of Mexico and is a comprehensive study of the factors influencing groundwater recharge in the semi-arid region. The text is well organised, adequately to the content, and supported with valuable supplementary material. The research approach using multivariate statistics is appropriate and gives a solid basis to conclusions. A disputable thing is a large number of self-citations (J.A. Ramos Leal cited seven times). However, it is less than 7% of the references and does not significantly influence the paper's quality.

Author Response

Dear Reviewer 3,

Thank you very much for your comments. We would like to mention that the fact of citing J.A. Ramos-Leal seven times is due this author has been one of the precursors in terms of hydrogeological studies in the region. Thus, we consider his works contributes with valuable information in this manuscript.

Round 2

Reviewer 2 Report

  1. I don't think the abstract summarizes the conclusion well. It must be improved.
  2. The deep discussion is suggested to supplement.

Author Response

Comments and Suggestions

The changes requested are colored yellow

Answers

The changes made are colored green

1.     I don't think the abstract summarizes the conclusion well. It must be improved.

Thanks you very much for your recommendation. We improved the abstract.

Identifying contributing factors of potential recharge zones is essential for sustainable groundwater resources management in arid regions. In this study, a data matrix with 66 observations of climatic, hydrogeological, morphological, and land use variables was analyzed. The dominant factors in groundwater recharge process and potential recharge zones were evaluated using K-means clustering, Principal Component Analysis (PCA) and geostatistical analysis. The study highlights the importance of multivariate methods coupled with geospatial analysis to identify the main factors contributing to recharge processes and delineate potential groundwater recharge areas. Potential recharge zones were defined into cluster 1 and cluster 3; these were classified as low potential for recharge. Cluster 2 was classified with high potential for groundwater recharge. Cluster 1 is located on a flat land surface with nearby faults and it is mostly composed of ignimbrites and volcanic rocks of low hydraulic conductivity (K). Cluster 2 is located on a flat lowland agricultural area, and it is mainly composed of alluvium that contributes to a higher hydraulic conductivity. Cluster 3 is located on steep slopes with nearby faults and is formed of rhyolite and ignimbrite with interbedded layers of volcanic rocks of low hydraulic conductivity. PCA disclosed that groundwater recharge processes are controlled by geology, K, temperature, precipitation, potential evapotranspiration (PET), humidity, and land use. Infiltration processes are restricted by low hydraulic conductivity, as well as ignimbrites and volcanic rocks of low porosity. This study demonstrates that given the climatic and geological conditions found in the Sierra de San Miguelito Volcanic Complex (SSMVC), this region is not working optimally as a water recharge zone towards the deep aquifer of the San Luis Potosí Valley (SLPV). This methodology will be useful for water resource managers to develop strategies to identify and define priority recharge areas with greater certainty.

(Lines: 21-41)

2. The deep discussion is suggested to supplement.

Thanks a lot for your suggestion. We have supplemented the discussion. The changes are shown in the next lines:

Similar studies have documented that the changes in climatic conditions such as, increase in temperatures are associated to lead to greater soil humidity deficits. This can decrease the magnitude of groundwater recharge affecting water availability [90].

(Lines: 262-265)

Similar studies have considered largely geologic features to identify recharge potential zones [111]. This is due to the recharge involves complex interplay factors, during infiltration process from vadose zone and saturate zone.

(Lines:310-312)

Previous studies have reported that the recharge depends on the nature and hydraulic properties of the vadose zone. This is due to the unsaturated layers are often defined by the permeability and porosity features which vary from point to another. Thus, groundwater recharge is governed by subsurface geology. In addition, the occurrence of fractures and faults are linked to increase of water movement (high hydraulic conductivity) to saturated zone [111]. However, variation of natural recharge process associated to PET and temperature, is due to the groundwater recharge is con-trolled, in part, by precipitation and evaporation, itself dependent on temperature [124,125]. A high association between humidity and land use can be due to a large area of agricultural fields in the study zone. Studies have demonstrated that the zones with high groundwater recharge potential are ubicated in areas with dense vegetation cover coupled with flat relief and consolidated and structured soils [126].

(Lines:464-475)

In previous studies, researchers have highlighted that suitable slope and elevation de-lineated by basin and depressions are crucial on recharge. This is due to the runoff is larger over zones of steeper slopes making no time for infiltration. Thus, it is probable that recharge is decreased as runoff flows faster and would not allow infiltration [109,127,128].

(Lines:482-486)

It is well known that the temperature and precipitation can have meaningful effects on groundwater recharge [129,130]. Particularly the recharge is significant in semi-arid regions where rainfall is variable and evapotranspiration frequently exceeds precipitation [109,131]. In addition, previous studies have found a high association between recharge and both climate and land cover [125]. Some researchers have documented that, variations in groundwater recharge are strongly related to vegetation types [129].

(Lines:495-500)
